# Distribution of coastal high water level during extreme events around the UK and Irish coasts.

Julia Rulent[1,2], Lucy M. Bricheno[2], Mattias J.A. Green[1], Ivan D. Haigh[3], and Huw Lewis[4]

[1]School of Ocean Sciences, College of Environmental Sciences and Engineering, Bangor University, Menai Bridge, United Kingdom
[2]National Oceanography Centre, Liverpool, United Kingdom
[3]School of Ocean and Earth Science, National Oceanography Centre, University of Southampton, Southampton, United Kingdom
[4]Met Office, Exeter, United Kingdom.

**Correspondence:** Julia Rulent (jrule@noc.ac.uk)

**Abstract.** The interaction between waves, surges and astronomical tides can lead to high coastal total water level (TWL), which can in turn trigger coastal flooding. Here, a high resolution (1.5km) simulation from a UK-focused regional coupled environmental prediction system is used to investigate the extreme events of winter 2013/4 around the UK and Irish coasts. The aim is to analyse the spatial distribution of coastal TWL and its components during this period by assessing 1- the relative contribution of different TWL components around the coast, 2- how extreme waves, surges and tide interacted and if they occurred simultaneously 3- if this has implications in defining the severity of coastal hazard conditions. The TWL components' coastal distribution in winter 2013/4 was not constant in space, impacting differently over different regions. High (>90[th] percentile) waves and high surges occurred simultaneously at any tidal stage, including high tide (7.7% of cases), but more often over the flood tide. During periods of high flood risk a hazard proxy, defined as the sum of the sea surface height and half the significant wave height, at least doubled from average over ¾ of the coast. These results have important implications for the risk management sector.

## 1 Introduction

High total water levels (TWL), arising from combinations of waves, storms surges and tides can lead to dangerous coastal conditions (Idier et al. (2019); Vousdoukas et al. (2018); Wolf (2009)). In the UK, coastal flooding is the most threatening natural hazard for coastal communities and environment (Home Office (2017)). In 2013, it was assessed that 520,000 properties in the UK (about 70% of which were homes) were in locations where the annual risk from coastal flooding was of 0.5% or greater (Committee on Climate Change (2013)). The financial implications of flooding can be severe: during the exceptionally stormy winter of 2013/4 coastal flooding in the UK was estimated to cause damage amounting to a value of £592.1 million (Chartteron et al. (2016)). In Ireland, the extreme conditions of that winter lead to high coastal erosion rates with extreme environmental impact (Sánchez-Arcilla et al. (2016); Cox et al. (2018); Janjić et al. (2018)) and flooding, especially over the south, west, and north-west coasts (Met Éireann (2014); Thorne (2014));insurance claims and repairs resulting from the

extreme weather damages amounted to more than €210 millions (Kandrot et al. (2016)). One problem is that storms and floods can vary in both magnitude and distribution (Kirezci et al. (2020)), making mitigation costly and difficult. For example, storm Xaver, and its associated storm surge, generated higher still water levels than ever recorded before at multiple tide gauges (TGs) around the UK coast (Spencer et al. (2015)) on the $5^{th} - 6^{th}$ December 2013. That same winter, storm Anne occurred between the $3^{rd} - 4^{th}$ January 2014 (RMS, 2014), leading to widespread flooding along southern UK coastal regions (Haigh et al. (2015); Sibley et al. (2015)). This was as a result of extreme TWL through the coincidence between the surge and spring tides (RMS (2014); Sibley et al. (2015)). Storm Ulla, also known as the 'Valentine Day Storm' as it occurred on the $14^{th}$ February 2014, hit the UK with extreme winds (RMS (2014)) and lead to some of the largest recorded skew surges over several sites in south-western UK (Haigh et al. (2017)). The frequency of these kind of event is expected to increase in the future (Stocker et al. (2013)), which matters not only because the number of individual storms could augment, but also because their impact is increased by the clustering of events (Priestley et al. (2017)). During winter 2013 the frequency of intense cyclones was almost twice that of the climatological average (Priestley et al. (2017)), and the proximity of events increased the impact of the storms over the British Isles (Priestley et al. (2017)). This, together with other factors such as the compound risk from coastal TWL and high river discharge (Khanal et al. (2019); Moftakhari et al. (2019)) or the urbanisation of coastal regions Stevens et al. (2016), may increase threats from coastal flooding (De Dominicis et al. (2020); Horsburgh et al. (2020); Stocker et al. (2013)). It is therefore important to understand how the sea level components behave at the coast and interact during extreme events.

There are several processes to consider in the tide-surge, tide-waves and surge-waves interactions. For example, both surges and tide act as shallow water waves and can modulate each other's phase leading to extreme surges to often occur over the rising tide (Horsburgh and Wilson (2007)). The interaction between non-tidal residuals and the tide can reduce extreme sea level by up to 30% (Arns et al. (2020)) and is therefore a key parameter to consider when studying coastal high water level formation. Extreme waves are also a crucial component of TWL and can affect (and increase) surges by altering the surface roughness (Bertin et al. (2015); Idier et al. (2019)). The dependence between waves and surges can double the 1 in 100 year extreme water level return period and lead to extreme coastal conditions (Marcos et al. (2019)). It is worth noting that this type of extreme value statistics can be altered by new extreme events; storm Xaver, occured at the beginning of December 2013, has increased the high water level estimates for a 1 in 200 years return period of up to 0.4m in some areas of the German Bight (Dangendorf et al. (2016)). Moreover, waves can be affected by tidal currents (Ardhuin et al. (2012)) and by tidal water levels, with high tide and deep waters allowing waves to travel further, with less dissipation at the seabed. At low tide, this same modulation of water depth increases the bottom drag and creates a larger sink of wave energy. Excluding dissipation processes, waves travelling against the ebb current direction can increase in height and steepness (Idier et al. (2019)). When these processes combine they can lead to high coastal TWL that may be dangerous for certain areas.

In this paper we study the interactions between TWL components around the UK and Irish coasts. The focus is on offshore conditions, up to 1.5km from land with a minimum depth of 10 m, to investigate where high-water levels can occur before waves enter the surf zone. We selected the winter 2013/4 as a case study, since it was an exceptionally stormy season, to investigate the coastal conditions during the extreme events of this period and the processes that lead to it. The overall aim

is to analyse the spatial distribution of TWL along the UK coast during winter 2013/4, assessing the relative contributions of its major components: waves, tides, and surges. Where are the maximum waves and surges occurring during that period? Are extreme waves and surges concurring more often over specific regions? How was the coastal hazard risk affected by the conditions of that winter?

The methodologies and the extreme period selected is described in section 2. Results can be found in section 3 and are discussed in section 4. The final conclusions are found in section 5.

## 2 METHODOLOGY

The winter of 2013/4 is renowned for being a particularly extreme winter in terms of TWLs (Wadey et al. (2014)) and is therefore an interesting period to investigate. To set the context of conditions during winter 2013/4, we first apply the peak over threshold (POT) method (Coles (2001)) to a long-term wave model climatology. The 90[th] percentile of the significant wave height (Hs) from December 2013, January and February 2014, is compared to the 90[th] percentile of a 37 year (1979 to 2015) climatological run of the Wave Watch III model (WW3;Tolman and Iii (2014)), at 1/12° (approximately 9 km) resolution. We repeated this analysis to evaluate the relative magnitude of the surges during this period. The 90[th] percentile of the surge during winter 2013/4 was compared to that of a 27 year dataset from 1992 to 2019 using a combination of the tide-surge continental shelf models CS3 (Flather (2000)) and CS3X (Williams and Horsburgh (2013)) simulations, both with a resolution of 1/9° in latitude and 1/6° in longitude (approximately 12 km).

To analyse the spatial distribution of TWL components during extreme events in 2013/4, simulations from the high resolution wave-ocean-atmosphere regional coupled model UKC4 (Lewis et al. (2019b, a, 2018)) was used. This is a state-of-the-art coupled model which integrates WW3, the ocean model Nucleus for European Modelling of the Ocean (NEMO; Madec (2008)) and the atmospheric Unified Model (UM; Cullen (1993)). This model also allows the coupling to a river model, however in this configuration climatological river inputs are used instead. The UKC4 coupled system is aimed at replacing parametrisations with direct coupling, to more explicitly simulate feedbacks between waves, ocean and the atmosphere at scales that are relevant to coastal interactions. The ocean component (NEMO) uses a regular high resolution (1.5 km) grid, initialized from an operational hindcast simulation of the Atlantic Margin Model (AMM15; Graham et al. (2018)). The daily lateral boundary conditions are obtained from the operational 12 km resolution NAT12 North Atlantic ocean model configuration (Siddorn et al. (2016)). The atmosphere component (UM) overlaps with the ocean grid over the shelf region at 1.5 km resolution, increasing to 4 km resolution at the domain's edges. It is initialised on the 30[th] October 2013 from a global MetUM operational simulation at 25 km resolution (Lewis et al. (2019a)). The Wave component (WW3) run on a two-tier Multiple Cell grid (Li (2011)) varying from a resolution of approximately 3 km in water deeper than 40 m (open waters) to 1.5 km resolution in coastal regions. It is initialised by a 10 day uncoupled simulation of the wave model (WW3), while the lateral boundary conditions are provided hourly from a hindcast wave-only simulation of the same model (Lewis et al. (2019a)). There is a two-way hourly coupling set up between all of the system's component; this is achieved through the OASIS_MCT libraries (Valcke et al. (2015)). A more in depth description of the model set up can be found in (Lewis et al. (2019b, a, 2018); Valiente et al. (2021)). The model in

its fully coupled configuration is applied from the 01/12/2013 to the 28/02/2014, outputting hourly data at each point (see Fig 1 for the domain). Here, we focus on the coast, and consequently used the first grid cell from every land point in the domain (Fig. 1). Note that because the resolution is 1.5 km this area is as close to the coast as possible, however some nearshore effects will not be represented in the model. Shallow water dynamics as well as the tidal asymmetry and modulation of high waters in intertidal estuaries (Nidzieko (2010)) could lead to changes in the timing of waves, surges and tide inshore which cannot be represented here. For example, the model will not simulate flooding and drying, and wind waves represent an 'offshore' condition without shallow water transformation. The modelled Hs and sea surface height (SSH), including astronomical tides and surges, are extracted from the fully coupled simulation. To derive the surge from the SSH, information on the tide's behaviour without the influence of the atmospheric component is needed. This baseline signal is obtained from a tide-only simulation of the UKC4 over the same period as the fully coupled run. Data are used to calculate the non-tidal residual by subtracting the baseline from the SSH.

The model predictions were compared against observations for validation; the SSH is compared to values from 25 TGs located around the UK coast during the period between 01/12/2013 and 28/02/2014. These data are obtained from the British Oceanographic Data Centre (BODC; see data availability section). The model baseline tide is compared to the tide obtained from harmonic analysis of the TGs during the same period. Results from this comparison are shown in the supplementary material (Appendix A, Tables A1 and B1). For the TWL comparison, the absolute difference between model and observation is 0.31 m, the RMSE 0.37 m, and the correlation R is 0.98. For the tide comparison the absolute mean difference is 0.30 m, the RMSE is 0.35 m, and the correlation 0.99.

A key goal was to investigate the different contributions to coastal extreme water levels caused by waves, surges, and tides. To do this, for each coastal point, individual storms were first identified, by separating high-wave events where there is more than 12 hours between successive peaks. Once these separate events were identified, the most intense storms were selected by ranking the Hs peaks and considering the 10 highest Hs values. The 24-hour interval around each peak is then considered to represent a storm period. During these storm periods, we select the maximum Hs, surge, and tide value at each point in space, independently from the time they occurred on. The objective is to show which area was impacted more severely by each component.

To understand how the TWL components are distributed in time with respect to each other, we estimated the amount of extreme Hs and surges occurring over each hour of the tidal cycle (see the schematic in Fig 2); this is expressed as a percentage of the total number of extreme Hs and surge occurred during winter 2013/4. To do this, the first step was to calculate the 90th percentile of Hs and surges over the entire winter. During the storm periods chosen, all the hourly model outputs higher than the 90th percentile value were selected. We then found the closest tidal peak to each of these outputs, to investigate which tidal stage the extreme Hs or surge occurred on, from six hours before to six hours after high tide. The number of high Hs and surges occurring over each stage of the tide were counted. This number was converted to a percentage of the total number of individual high Hs and surges or simultaneous Hs and surges considered overall at each point. For example, if 10 extreme Hs occur at one point at any time during winter 2013/4, and 2 of these are occurring at high tide, we show that 20% of extreme Hs occurred at high tide. The mean of this percentage at all coastal points is calculated for each hour of the tidal cycle with

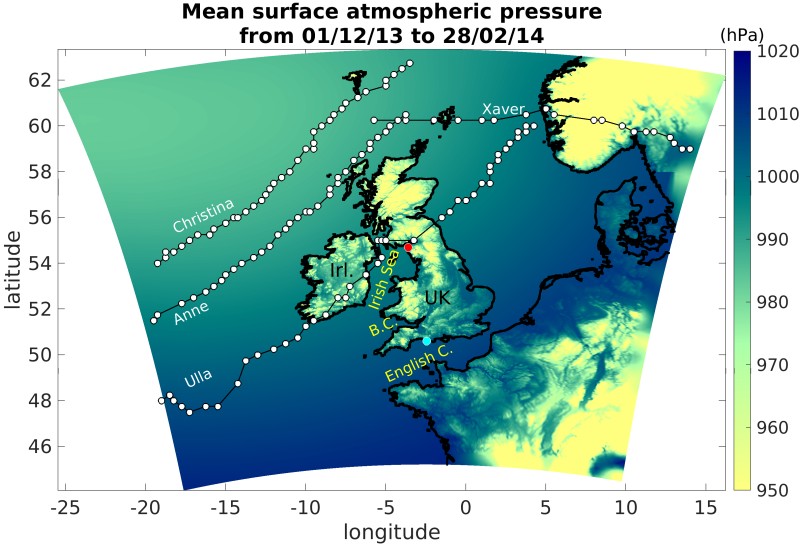

**Figure 1.** The UKC4 model domain with shading giving the UKC4 simulated mean surface atmospheric pressure during the 3 months of winter 2013/4. The figure also shows the estimated track of four storms during winter 2013 (storms Xaver 5/12/2013, Anne 3/01/2014, Christina 7/01/2014, and Ulla 14/02/2014). Tracks are obtained from the mean sea level pressure (MSLP) of hourly ERA5 reanalysis data. The white markers show the location of hourly minimum MSLP during the storm period. The red and blue dots show the locations of Solway Firth and Weymouth respectively, which are mentioned in the discussion.

the respective standard deviation (SD). Note that 'simultaneous' refers to high Hs and surges data outputted during the same hour. We also show the spatial distribution of the number of simultaneous extreme Hs and surges at high tide, as well as the differences between the latter and the values for low tide, three hours before high tide, and three hours after high tide. This allows us to better understand the interactions between extreme Hs, surges, and tides during winter 2013/4.

To understand how the joint contribution of waves, surges, and tides affected the coastal hazard under these conditions, a hazard proxy (HP) was defined as SSH + 1/2Hs and this was calculated for each coastal point. This HP has been used in previous publications (e.g.,Lyddon et al. (2019)) under the premise that operationally flood warnings are issued when predefined thresholds, based on water level and waves, exceed a specific level (Del Río et al. (2012); Lawless et al. (2016)). While different regions have a different threshold for operational purposes as they are, this proxy is used in this study to provide a national-scale picture to visualise how the severity of conditions varies around the UK coast. The HP is calculated for all time in which the tide is higher than the lowest high water, similar to the approach of (Lyddon et al. (2019)), to focus on conditions in which overtopping is most likely.

## 3 RESULTS

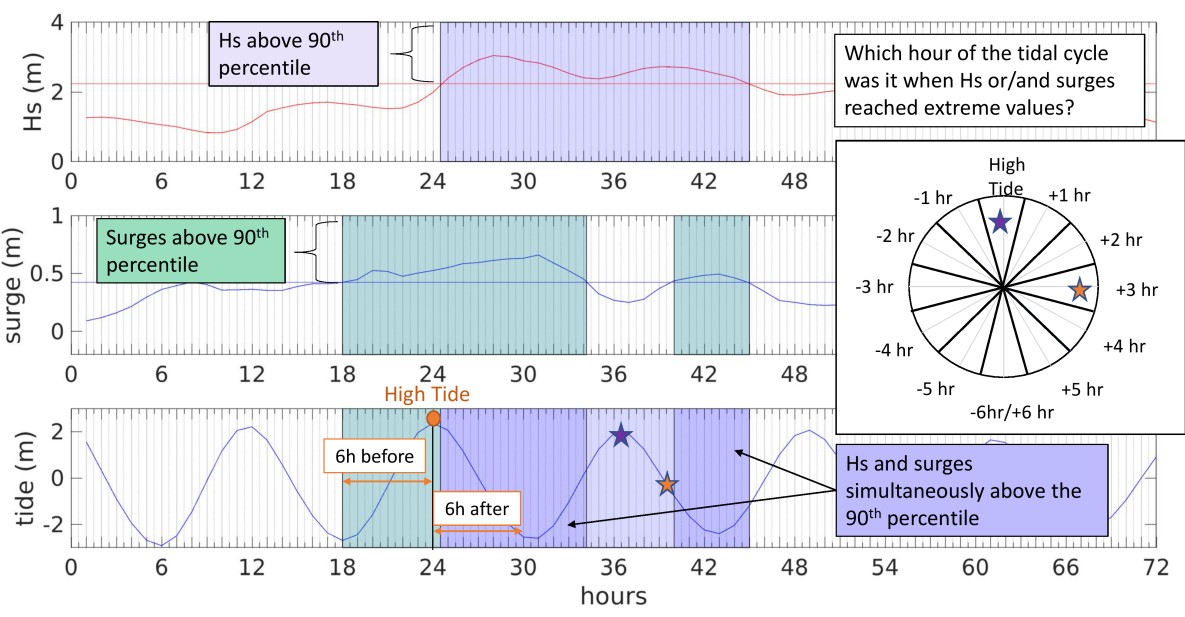

**Figure 2.** Schematic example explaining the methodology used to evaluate the amount of extreme Hs and surges with respect to the tidal cycle. This example shows three day of the time series at one coastal point. The Hs values higher than the 90th percentile (calculated over 3 months of data at the same point) are considered high. The closest tidal peak to each value is found. We then evaluate when each value occurred in the tidal cycle, from 6h before to 6h after high tide. The same is done for surges, and for cases in which Hs and surges are simultaneously higher than the respective 90[th] percentile. The two stars indicate examples, where the extreme Hs occurred at high tide, and three hours after high tide.

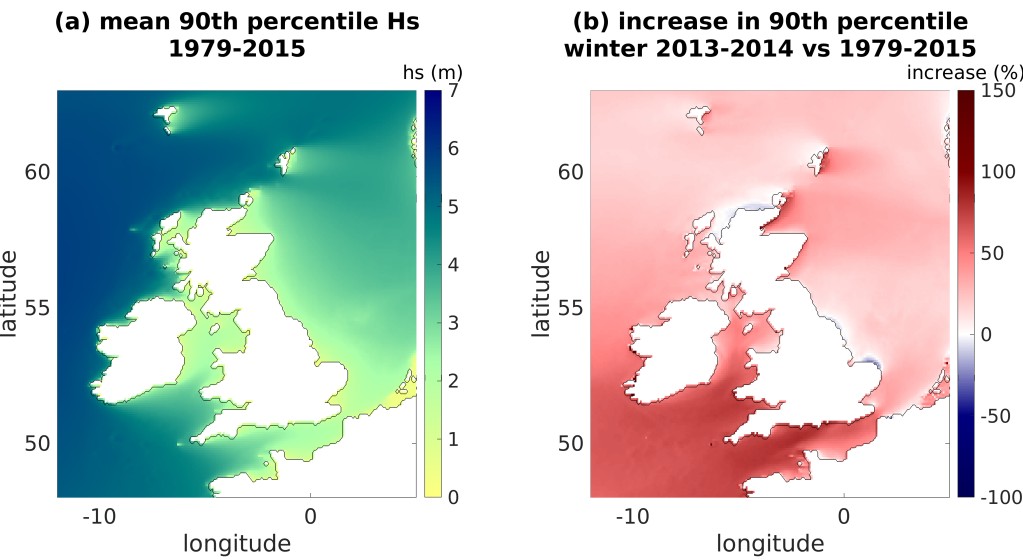

**Figure 3.** Mean of the Hs's 90 <sup>th</sup> percentile from the climatology run (a). Percentage increase in Hs in winter 2013/4 compared to climatology (1979-2015) derived from WW3 simulation(b). Note the skewed colour range and that in some areas of the UK east coast and north of Scotland, the Hs in fig 3 does reduce.

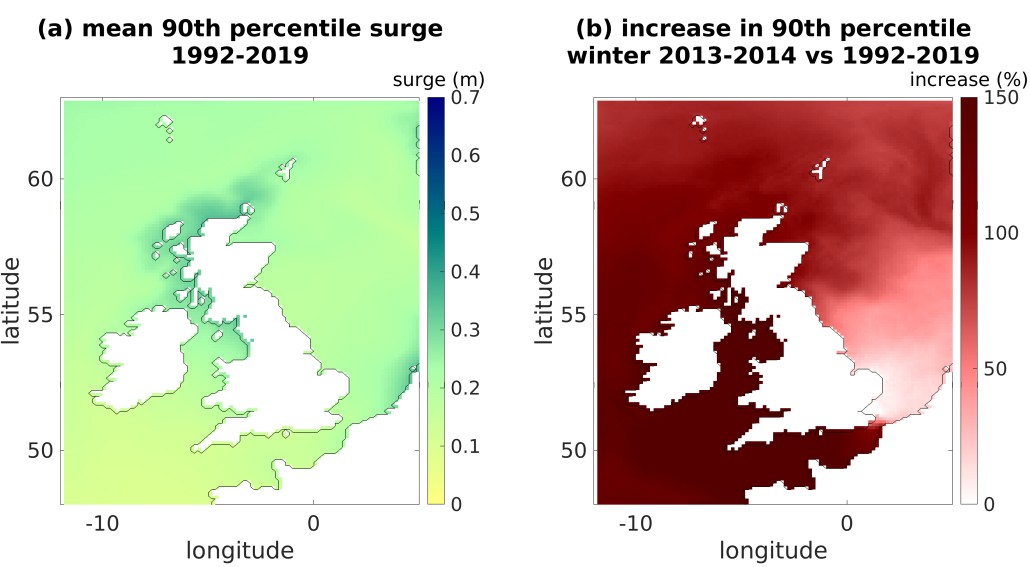

**Figure 4.** Mean of the Surge 90 [th] percentile from the climatology run (a). Percentage increase in surge in winter 2013/4 compared to climatology data (1992-20019) from CS3 (1992-2006) and CS3X (2007-2019) simulations (b). Note that the colour scales are different between figures 3-4.

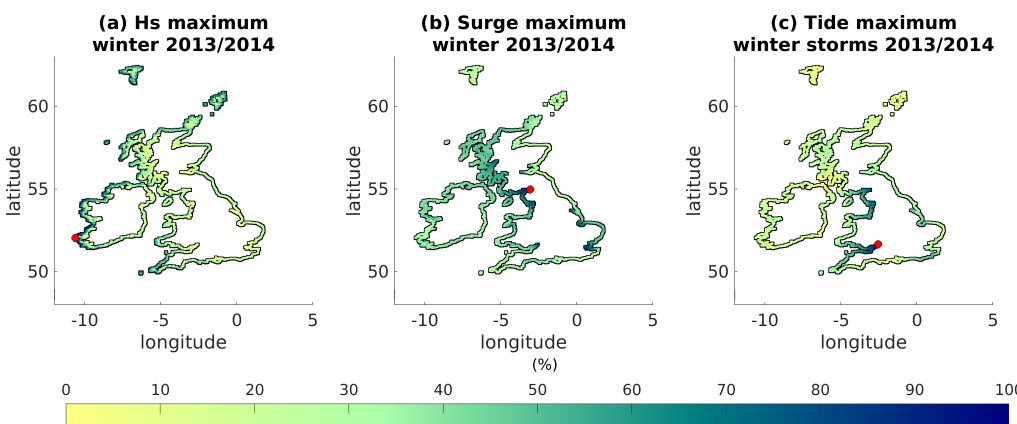

**Figure 5.** Spatial distribution of the UKC4 simulated maximum Hs (a), Surge (b) and tide (c) that occurred during the winter 2013/4. Maxima are given for each coastal point independently of when they occur. Values are expressed as a percentage of the respective overall maximum found over the entire coast in winter 2013/4: these are 14.1 m for Hs, 2.7 m for surges and 8 m for the tide. They are indicated by the red dot. Note that values in some areas seemingly change as distance from the coast (eg. Hs in the West of Ireland), this is because the coastal geomorphology is quite complex in those regions and the high resolution of the model leads to values changing drastically over short spaces. Only the closest grid cell to the coast is used.

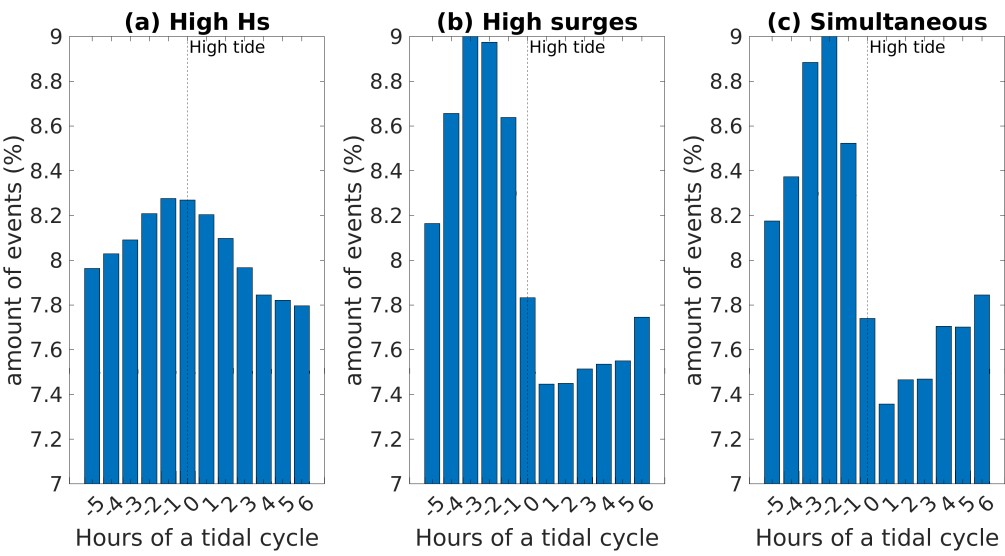

**Figure 6.** Percentage of the high (>90 [th] percentile) Hs (a) and surges (b) occurring as a function of tidal cycle stage during winter 2013/4 storm events. Percentage of simultaneous events shows when both Hs and surges are within upper 90th percentile concurrently (c).

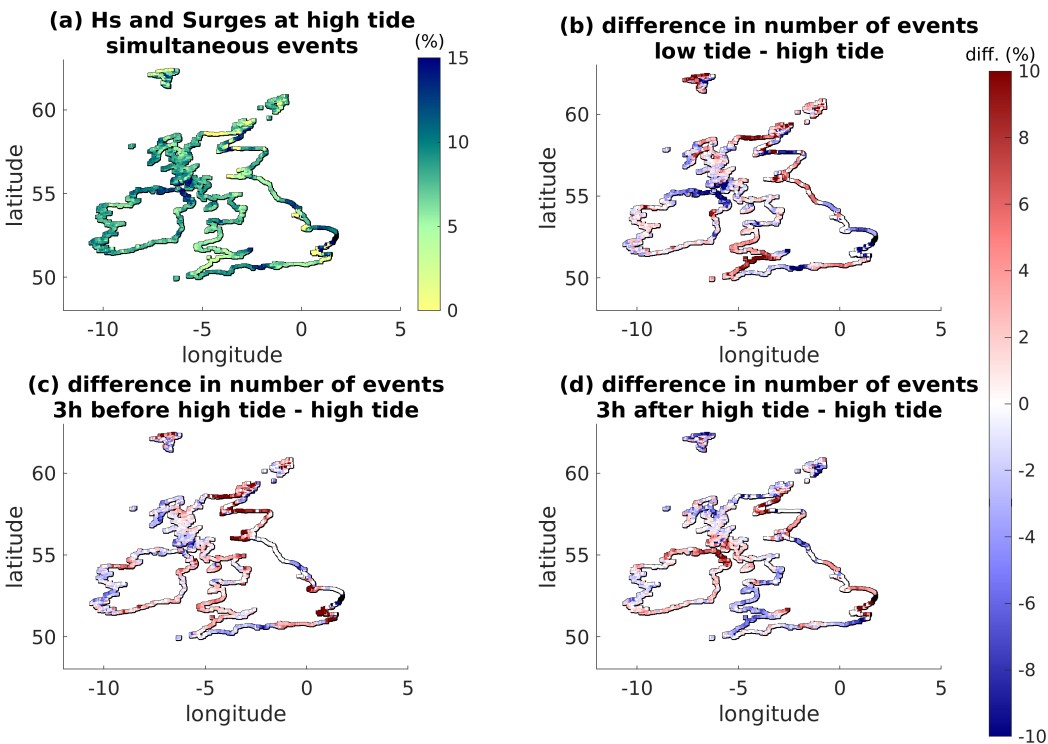

**Figure 7.** Number of hours in which high (above 90th percentile) Hs and Surge are concurring within one hour of high tide (a). This number is shown as a percentage of the total number of simultaneous events recorded during the winter at each location. The plot also shows the differences between the percentage of events occurring at high tide and those occurring at low tide (b), three hours before high tide (c) and three hours after high tide (d). This shows how the simultaneous events are distributed over the tidal cycle.

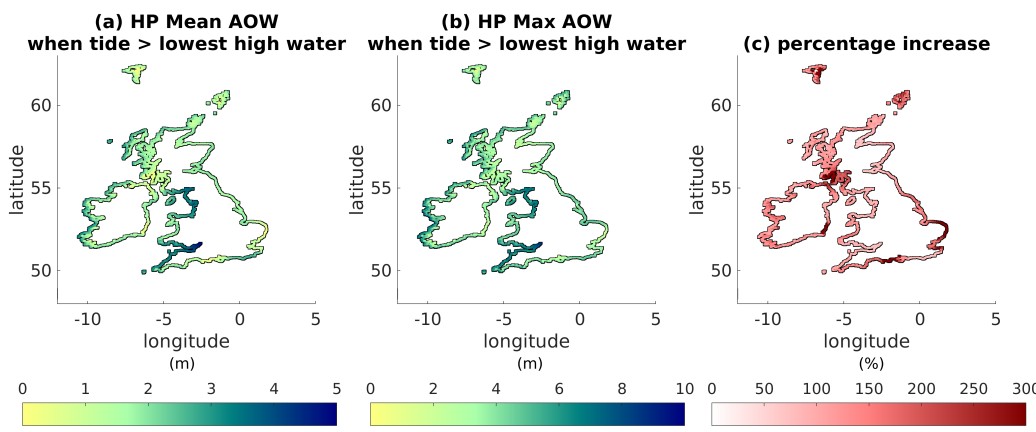

**Figure 8.** Mean and maximum hazard proxy calculated at each coastal point (a, b), and percentage increase between the mean and maximum (c). Note that the percentage increase plot was saturated to zero to only focus the positive increase, some regions of the Faroe Islands showed a decrease. Also note that this is the mean of values for winter 2013/4 which is an extreme period, the comparison to a longer climatology simulation could show a higher increase.

Our results show that during winter 2013/4 extreme wave and surge events are higher than in a typical year, with the Hs 90[th] percentile of that period on average 35% higher than the 37 year climatology from 1979 to 2015 (Fig 3) and the surge 90[th] percentile on average 106% higher than that of the period from 1992 to 2019 (Fig 4).

To study this extreme period and the conditions that characterised it, the waves, surges, and tide simulated in the UKC4 regional coupled system are considered separately. The spatial distribution of their maximum values (Fig 5) shows that each component of the TWL is not evenly distributed around the UK coast. Highest Hs can be seen along stretches of coastlines exposed to the Atlantic, including the South-West and North-West UK, and the West coast of Ireland, where Hs reached up to 14.1 m. The highest surges, reaching up to 2.7 m in Solway Firth, impacted the west coast of England and Scotland, especially in the North-East Irish sea and in the Bristol channel. They were also dominant along the South-East coast of England. Independent of the storm events, astronomical tides are larger in the eastern Irish Sea, in the English Channel, and in the Bristol Channel, where they reach up to 8 m. These maxima are given independently from the time at which they occur and therefore show which regions are affected more by each component, but do not show if and how these extremes interacted together.

To better understand the interaction between the water level components, the timing of high Hs, surge and tide is analysed (Figs 6 and Fig 7). The percentage of extreme Hs and surges occurring over each hour of the tidal cycle is calculated relative to the total number of extreme Hs and surges during winter 2013/4. Results show that high Hs occurred more often near high tide with up to 8.3% (SD 1.5% ) of events one hour before the high-water peak, while the high surges were found more often half-way through the flood tide, with 9% (SD 4% and SD 3.4%) of events occurring respectively three and two hours before high tide, followed by 8.7% (SD 2.4% ) of events four hours before high tide (Fig 6). There is a clear pattern in both Hs and surges curves (Fig 6), with more extreme Hs and surges occurring over the flood tide than over the rest of the tidal cycle, although this signal is more pronounced in surges than waves. The pattern of the curve representing simultaneous Hs and surges is a combination of those signals, with a dominant influence from the surges. Simultaneous events occurred over each stage of the tidal cycle, but more often during the flood tide, with 9% (SD 6.6% ) and 8.9% (SD 5.1% ) of events occurring two and three hours before high tide. High Hs and surges also occurred at high tide, with 8.3% (SD 1.5% ) of Hs, 7.8% (SD 2.2% ) of surges and 7.7% (SD 3.6% ) of the simultaneous events happening within the hour of high tide. Looking at the distribution of events at the coast (Fig 7) some areas also have an increase in simultaneous events throughout the ebb tide. A higher number of simultaneous events is recorded over the western and southern coast of UK, with up to 21 hours of concurring Hs and surges recorded in proximity of Weymouth two hours after high tide. In the same region 20 hours of simultaneous extreme Hs and surges are also recorded over the flood tide. These are not necessarily consecutive hours and could indicate multiple storms hitting the area. During the period of coincidence between extreme waves, extreme surges, and high tides the regions showing the highest TWLs are over the west of Ireland, Scotland and in the Bristol Channel, followed by the north-west and south-east of England.

The implications of these simultaneous events can be quantified in terms of a coastal hazard mean and maximum HP value for each region of the coast during periods in which overtopping is considered more likely (Fig 8). This value, similarly to the TWL, represents the simultaneous contribution to the water level from waves, surges and tides. However, operational

thresholds are not based on just the maximum values of the TWL but are set considering extreme conditions, therefore the HP is calculated from the SSH (including astronomical tides and surges) summed to only half of the Hs rather than the maximum.

The calculation shows that the mean HP over all coastal points in the model domain is 1.8 m, ranging from -0.18 m in the Faroe Islands to 4.9 m in the Bristol Channel. The maxima in HPs are found at the west coast of Scotland, Ireland, North west of England, and the Bristol Channel, where values reach 8.8 m in areas where the average HP is of 4.3 m. The maximum HP is above 3m over more than 80% of the coast. This includes regions of the east coast of UK that are affected by a lower number of extreme Hs and surges compared to the west and south coast. The increase in the HP from the average value to the maximum

recorded that winter show that the water level during the most threatening times more than doubled over 75% of the British and Irish coasts combined.

## 4 DISCUSSION

From a high resolution regional coupled model simulation of winter 2013/4 it is possible to analyse the dominant factors driving high coastal TWL around the UK. It is important to underline that this study does not show a return period of events

and cannot show which areas of the UK have higher flood risk. Rather, it provides a case study of extreme conditions to better understand the processes related to the formation of high coastal TWLs. Considering the maximum Hs, surges, and tides individually (Fig 5), we showed that some areas of the UK are dominated by specific components of the TWL during winter 2013/4. The climatology run compared to the maximum surges and waves shows that the highest residuals are distributed over the North-West and South-West coast of England, while the highest Hs are distributed over the regions open to the Atlantic

Ocean. Therefore, different types of hazards affected each area. This implies that in a changing climate, it is important not only to consider the TWL overall but the different hazards coming from each component. Regions that have been at risk because of strong winds with wave breakers set up as defences might become affected by surges or tides which require a different approach. During winter 2013/4, a sequence of extreme storms induced by an unusually strong North Atlantic jet stream (RMS (2014)) followed a more southerly path than usual (Thorne (2014)), causing extended flooding over the north of Scotland, west

of Wales, west, south-west and south-east of England (Haigh et al. (2016); RMS (2014); Thorne (2014)). The extent of storms over that season lead to question if the ongoing changes in the climate and typical storms conditions may also lead to changes in the spatial distribution and duration of floods (Thorne (2014)).

The interaction between Hs, surges and tides is the main driver of high TWL. If the maximum waves, surges and tide (fig 5) could occur simultaneously, the highest TWLs would reach above 16 m in areas of the west of Ireland 13 m in areas of the

200 Bristol channel. It is therefore important to understand if extreme values from each component could happen simultaneously and, if not, which could be the worst-case scenarios. A previous study shows that most extreme sea level events are strongly dependent on the tidal stage over which they occur (Haigh et al. (2016)). During winter 2013/4, high (more than $90^{th}$ percentile) surges occurred more often halfway through the rising tide, 3-4 hours before high tide (Fig 6). This is consistent with previous observational studies finding that the peak surges tend to appear 3-5 hours before high tide (Horsburgh and Wilson (2007)).

This is because surge and tide, behaving like two shallow water waves, mutually affect each other's phase as a function of water depth (Horsburgh and Wilson (2007); Rossiter (1961)).

    The results in Fig 6 also show a higher percentage of extreme waves occurring in the hours before high tide. It should be noted that the coastal region in the model domain is defined as the first 1.5 km from land, which has a minimum depth set at 10 m. Wind waves in this area are all deep-water waves (or in some rare cases intermediate-water waves). The coastal water depth

does not impact directly on them, but changes in tidal currents, wind and low-pressure system will affect them. Resolving the surf-zone processes, essential for operational flood-hazard studies, would require higher resolution model studies. However, our results show accurate near-shore conditions that could be used as input to drive higher-resolution flood models for further studies of specific regions. Moreover, our study did not investigate the magnitude of waves and surges per se, but the recurrence of extreme values over the tidal cycle. Other studies focusing on the period January-February 2014 show that, in some regions

of the Irish sea, the tide-wave interaction, including tidal currents induced modulation of wave refraction, can lead to an increase in the magnitude of Hs (Lewis et al. (2019c)). Larger waves during high water were demonstrated to be up to 20% larger because of the interaction with the tide. In the context of our study, this means that in some cases extreme waves could not only occur more often near high tide, but also be higher during this period because of the very interaction with high waters.

    Due to a combination of the processes described above, simultaneous high Hs and surges occur more often two to three hours

before high tide and rarely at high water (Fig 7). From observational studies the residual peak has been shown to rarely occur within an hour of high tide (Horsburgh and Wilson (2007)), which is consistent with these coupled model results. However, results also show that it is possible for all three components of the TWL to reach peak values simultaneously in 7.7% of cases considered with a SD of 3.6% . This means that in most coastal points at least 4.1 % of extreme waves and surges concurred at high tide. The coupled model data also shows that 7.8% of events concurred at low tide (Fig 6 and Fig 7). In this case

simultaneous high surge and Hs are not a threat since they occur over low tide and no observed skew surge event during winter 2013/4 coincided with extreme sea level events at low tide (Haigh et al. (2016)).

    Observational studies linked extreme sea level events occurring along the south-western England coast with storms travelling to north of the UK (Dhoop and Mason (2018); Haigh et al. (2016)), with centres located west or north-west of Ireland, north of Scotland, or around Scandinavia (Haigh et al. (2016)). This is consistent with the atmospheric model outputs from this coupled

simulation (Fig 1) and with the majority of extreme simultaneous events being simulated over the west coast and south-west coast of UK (Fig 7). This area was affected by several flood events during that period (Haigh et al. (2016)) which underlines the importance of tide-wave-surge interactions. In areas such as the west coast of Ireland, where the highest Hs are recorded but high waves do not often concur with surges, the same TWL values can be obtained as for other areas with lower waves and simultaneous high surges or tides. Both cases can represent a different kind of threat for a given coastal area. These results

suggest the need for further analysis of the relative contribution to TWL from waves tides and surges in sub-sections of the coast to understand if it can be representative of specific regions, considering the influence of storm tracks and duration in more details.

    When considering periods during which overtopping is more likely, the HP reaches on average above 1.8 m across the UK and Irish coasts, but the maximum HP (Fig 8) reaches at least 3 m over more than 80% of the coastal region considered. The

HP reached above 6 m over the coast of the North east Irish Sea and the Bristol channel, where peak surges are dominant, as well as over the west of Ireland, where extreme waves are dominant. This shows that each of the TWL components should be considered when trying to assess coastal hazard since each of them can lead to hazardous conditions in particular circumstances. Note that this HP cannot give information as to whether a region is more at risk of flooding than another but gives an understanding of how the TWL distribution can change at the coast during extreme periods at risk of flooding. The operational

coastal risk levels are individual to each region, which will have different flood hazard threshold depending on local conditions (bathymetry, geomorphology etc.). However, results show that the maximum increase in HP from the mean at least doubled over three-quarters of the UK coastline. For a quarter of coastal points the HP values increased more than $160\%$. This is an increase from an average value calculate from an extreme winter, which means that the value will underestimate the increase from a typical winter condition.

In the future it would be interesting to consider individual storm events to focus on the importance of the storm tracks and storm timing over the distribution of coastal water levels, exploiting the coupling flexibility of the UKC4 system to study sensitivity to factors such as the atmospheric forcing. Moreover, observational studies showed that the average storm duration from climatological records can be very different over the east and west coast of the UK, with increased likelihood of high waves coinciding with high waters over the west coast where storms are on average longer (Dhoop and Mason (2018)). It

would be interesting to consider the duration of these events in future studies to understand if several smaller storms could lead to a similar threat as one longer storm.

## 5   Conclusions

In this paper we have undertaken an analysis of the coastal spatial distribution of TWL and its major components; waves, tides, and surges. A consistent analysis of these components is enabled by considering results of fully coupled wave-ocean-

atmosphere regional coupled model simulations. Results showed that during the extreme winter of 2013/4, the highest simulated waves impacted over the Atlantic coast, while surges were dominant over the north west coast of England, in the Bristol channel and south west of UK. The tide was dominant in the north west of England, Bristol channel and English Channel. Results show that each of these components individually can lead to high coastal water levels and should all be considered when assessing severe coastal conditions. The overall highest Hs, surge and tide are respectively 14.1 m, 2.7 m, and 8 m. During that

winter, extreme surges and Hs occurred individually and simultaneously over each stage of the tide. Most concurrent events are found two to three hours before high tide. There are more simultaneous high Hs and surges over the west and south-west coast due to the storm tracks of that period travelling north of the UK. However, the maximum HP can be significant everywhere in Britain. The HP during hazardous periods at least doubled over three-quarters of the coastal points in UK and Ireland. This suggests that other factors could have an important impact in determining the hazard of a specific regions, as for example

the duration and track of storms which should be investigated in future studies. Results also showed that during the extreme 2013/4 winter, it was possible for extreme Hs, surges, and tide to occur simultaneously leading to extremely dangerous coastal conditions.

*Data availability.* The nature of the 4-D data generated in running the various experiments requires a large tape storage facility to maintain the model data. These data amount to several tens of Tb archived on the Met Office MASS system. These data can be made available to

interested researchers by contacting the author, and will require signing of licence agreements in order to access the data. Each simulation namelist and input data are also archived under configuration management, hosted by the Met Office, and can be made available to others upon contacting the authors. The model was validate against TG from BODC, accessible at:

https://www.bodc.ac.uk/data/hosted_data_systems/sea_level/uk_tide_gauge_network/

## Appendix A: Appendix A

**Table A1.** Comparison of TWL to TGs from 01/12/2013 to the 28/02/2014. To assess against observations with a higher temporal frequency than model outputs, both the TG and the model data are interpolated to one-minute time series and then re-averaged within the hour. In some cases, we found an offset of 1 hour in the peak time from model and observation. The percentage of peaks misrepresented by the model is shown in the last column. In nearly all this cases the model peaks occur one hour earlier than in observations. the columns show the TG sites names, the mean absolute difference between the model minus the TG (MD), the root means square error (RMSE), the correlation (R), the mean peak magnitude difference (MPD), and the ammount of cases in wich the model peaks were offset in time compared to that of TGs, i.e. the peak time difference (PTD).

| TG Site | MD (m) | RMSE (m) | R | MPD (m) | PTD (%) |
|---|---|---|---|---|---|
| Aberdeen | 0.26 | 0.32 | 0.98 | -0.08 | 32.76 |
| Barmouth | 0.18 | 0.23 | 0.99 | -0.09 | 24.71 |
| Cromer | 0.48 | 0.53 | 0.99 | -0.24 | 5.75 |
| Devonport | 0.28 | 0.33 | 0.99 | -0.27 | 30.46 |
| Dover | 0.41 | 0.48 | 0.99 | -0.12 | 2.87 |
| Fishguard | 0.22 | 0.24 | 0.99 | -0.21 | 4.02 |
| Heysham | 0.47 | 0.62 | 0.97 | 0.26 | 7.47 |
| Hinkley | 0.27 | 0.34 | 0.99 | -0.18 | 14.37 |
| Holyhead | 0.22 | 0.25 | 0.99 | -0.23 | 14.94 |
| Ilfracombe | 0.28 | 0.34 | 0.99 | -0.32 | 13.79 |
| Kinlochbervie | 0.12 | 0.15 | 0.99 | -0.01 | 21.84 |
| Leith | 0.34 | 0.41 | 0.98 | -0.14 | 31.21 |
| Liverpool | 0.27 | 0.32 | 0.99 | -0.07 | 29.89 |
| LiverpoolTG | 0.27 | 0.32 | 0.99 | -0.07 | 29.89 |
| Llandudno | 0.20 | 0.25 | 1.00 | -0.06 | 6.90 |
| Lowestoft | 0.48 | 0.50 | 0.97 | -0.44 | 32.76 |
| Milford | 0.24 | 0.29 | 0.99 | -0.29 | 9.77 |
| Mumbles | 0.25 | 0.36 | 0.99 | -0.21 | 10.92 |
| Newhaven | 0.30 | 0.34 | 0.99 | -0.22 | 1.15 |
| Newlyn | 0.64 | 0.95 | 0.83 | -0.28 | 48.28 |
| Newport | 0.52 | 0.62 | 0.98 | 0.17 | 17.82 |
| StMarys | 0.32 | 0.35 | 0.99 | -0.31 | 3.45 |
| Tobermory | 0.13 | 0.16 | 0.99 | 0.00 | 18.97 |
| Ullapool | 0.13 | 0.16 | 0.99 | -0.03 | 20.11 |
| Whitby | 0.38 | 0.45 | 0.99 | -0.23 | 25.29 |
| mean total | 0.31 | 0.37 | 0.98 | -0.15 | 18.37 |

**Table B1.** Comparison of tide to TGs from 01/12/2013 to the 28/02/2014. To assess against observations with a higher temporal frequency than model outputs, both the TG and the model data are interpolated to one-minute time series and then re-averaged within the hour. In some cases, we found an offset of 1 hour in the peak time from model and observation. The percentage of peaks misrepresented by the model is shown in the last column. In nearly all this cases the model peaks occur one hour earlier than in observations. the columns show the TG sites names, the mean absolute difference between the model minus the TG (MD), the root means square error (RMSE), the correlation (R), the mean peak magnitude difference (MPD), and the ammount of cases in wich the model peaks were offset in time compared to that of TGs, i.e. the peak time difference (PTD).

| | | | | | |
|---|---|---|---|---|---|
| Aberdeen | 0.28 | 0.35 | 0.98 | -0.06 | 37.93 |
| Barmouth | 0.20 | 0.23 | 0.99 | -0.07 | 27.01 |
| Cromer | 0.15 | 0.19 | 0.99 | 0.12 | 8.05 |
| Devonport | 0.33 | 0.38 | 0.98 | -0.28 | 34.48 |
| Dover | 0.22 | 0.26 | 0.99 | 0.28 | 6.32 |
| Fishguard | 0.34 | 0.37 | 0.99 | -0.26 | 6.90 |
| Heysham | 0.50 | 0.65 | 0.96 | 0.39 | 2.30 |
| Hinkley | 0.26 | 0.32 | 0.99 | -0.03 | 12.07 |
| Holyhead | 0.35 | 0.39 | 0.99 | -0.29 | 22.99 |
| Ilfracombe | 0.31 | 0.37 | 0.99 | -0.24 | 16.09 |
| Kinlochbervie | 0.29 | 0.32 | 0.99 | -0.20 | 25.86 |
| Leith | 0.38 | 0.45 | 0.99 | -0.18 | 23.70 |
| Liverpool | 0.22 | 0.27 | 0.99 | 0.11 | 28.16 |
| LiverpoolTG | 0.22 | 0.27 | 0.99 | 0.11 | 27.59 |
| Llandudno | 0.26 | 0.31 | 1.00 | -0.05 | 12.64 |
| Lowestoft | 0.10 | 0.12 | 0.98 | -0.01 | 39.08 |
| Milford | 0.36 | 0.40 | 0.99 | -0.35 | 13.79 |
| Mumbles | 0.29 | 0.35 | 0.99 | -0.18 | 13.79 |
| Newhaven | 0.18 | 0.23 | 0.99 | 0.05 | 6.32 |
| Newlyn | 0.42 | 0.46 | 0.99 | -0.31 | 5.17 |
| Newport | 0.56 | 0.67 | 0.98 | 0.31 | 20.11 |
| StMarys | 0.40 | 0.43 | 0.99 | -0.34 | 6.90 |
| Tobermory | 0.32 | 0.35 | 0.99 | -0.21 | 33.33 |
| Ullapool | 0.29 | 0.33 | 0.99 | -0.19 | 31.61 |
| Whitby | 0.27 | 0.32 | 0.99 | -0.03 | 32.18 |
| mean total | 0.30 | 0.35 | 0.99 | -0.08 | 19.78 |

*Author contributions.*   JR,LB, and JAMG designed the research. JR lead the study, performed the research and the data analysis, and produced the figures. LB and JAMG help with the data analysis/interpretation and fundamental help in developing the manuscript and figures.HL provided the numerical model data and helped understanding all aspects related to the numerical model. IH helped with selecting the case study and understanding aspects related to surges and their impact during extreme events. HL and IH also help revising and improving the work and manuscript.

*Competing interests.*   They authors declare that they have no conflict of interest.

*Acknowledgements.*   this work used Monsoon2, a collaborative High-Performance Computing facility funded by the Met Office and the Natural Environment Research Council (NERC). This research has been carried out under national capability funding as part of a directed effort on UK Environmental Prediction, in collaboration between Centre for Ecology & Hydrology (CEH), the Met Office, National Oceanography Centre (NOC) and Plymouth Marine Laboratory (PML).

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
