# Peer review of "DISTRIBUTION OF COASTAL HIGH WATER LEVEL DURING EXTREME EVENTS AROUND THE UK AND IRISH COASTS."

_Natural Hazards and Earth System Sciences, 2021_

## Author Response (AR1)

ANSWERS TO COMMENTS FROM THE EDITOR AND THE REVIEWERS.

EDITOR

Comments to the Author:
Dear authors, thanks for submitting your replies to the reviewers comments. Based on the reviews, your replies and my own evaluation, the manuscript is returned for a major revision.
Please note that the decision for a major rather than a minor revision was taken so that I can re0send the manuscript again to at least one of the original reviewers
One more additional comment from my side: Priestley et al (2017) is a very nice synoptic description if the clustering of cyclones affecting the British Isles in the winter 2013-14, and I think it would be a nice addition to the literature review in the introduction.
Priestley MDK, et al (2017) The role of cyclone clustering on the stormy winter of 123/14. Weather 72:187-192 doi:10.1002/wea.3025
Looking forward to the revised version of your manuscript
Best regards, Joaquim Pinto (handling editor)

**Dear Editor,**
**Thank you for your comment. We are glad to hear that based on the reviews and your evaluation we can return the manuscript for a further revision.**

**The reference you suggested was added to the introduction along with a few sentences regarding storms clustering, P2L31-34. All the comments from the reviewers were taken into consideration and answered. We hope the resulting changes have improved the manuscript.**

**Please note that while reviewing the manuscript we have noticed an additional error, in P13L155. This is the result showing how often extreme surges occur over the tidal cycle. In the first manuscript we accidentally included the results that referred to the coast of the entire model domain, including parts of northern Europe, rather than just the British Isles. This change does not affect the discussion or conclusion. The change was from '9.1% (SD 3.3) and 9% (SD 4%) of events occurring respectively three and four hours before high tide' to '9% (SD 4% and SD 3.4%) of events occurring respectively three and two hours before high tide.' We apologise for the mistake.**

**We look forward to hearing what you think of the revised manuscript.**

**Best regards,**

**Julia**

REVIEWER 1

This paper presents an analysis of the distribution of total water levels (TWL) and its major components (waves, tides, and surges) along the British and Irish coast during the extreme winter 2013/2014. The authors used a regional coupled environmental prediction system with high spatial resolution (1.5 km) to obtain the necessary data. In general, the paper is well structured and well written. More information should be provided on the model setup. Some other minor concerns are summarized below.

1. P2L36 Might be also interesting to mention that an individual storm (in this case Xaver) is able to increase the water level estimates for a 1 in 200 years event by up to 40 cm (see Dangendorf et al., 2016, https://doi.org/10.1088/1748-9326/11/5/054001). This study is also interesting regarding the estimation of worst-case scenarios.

   **Thank you for this reference, it makes a very interesting point. This was added in P2L45-L48.**

2. P2L46 Delete 'coastal' at '... the coastal distribution ...' as you already say along the UK coast afterward.

   **Done.**

3. P2L53 Delete reference to Fig 1; one cannot see that 2013/2014 was an extreme winter from this Fig

   **Done.**

4. P3L55 Would be interesting to compare not only the 2013/2014 90th percentile values for Hs and surge but also if/where winter 2013/2014 exhibits maximum values for Hs and surge in comparison to the climatology

11. P14L163 Based on your results (having the climatology and the 2013/2014) case: can you make any assumption about worst-case scenarios.

15. Figure 5 How do Hs and surge maxima look like in the climatology? Would probably also show the benefit of the high resolution in UK4.

   **We thank you for raising these good points and comparing the maxima considering the climatology was something that we meant to originally include in paper. However, our aim was to only use the climatology to justify our selected case study and only refer to the high-resolution model for the science, to not make this work a comparison between models but rather focus on the outputs of the UKC4.**
   **The climatology run and the UKC4 have very different set ups and cannot be easily intercompared. To include a comparison with the climatology in our main discussion our options would be to 1- use the maximum Hs and surges values for winter 2013 coming from the same climatology run, which would make the discussion about a mix of two model, or 2- we could include the maximum from the UKC4 model to compare to the climatology, which would lead to a complex intercomparison between models.**

Ultimately the best solution would be to produce a long UKC4 run, but because it is quite computationally expensive it was not possible to do this yet. It would be great for a future study.
Having said this, we have included the following lines in the discussion about the maximum surge and Hs: 'The climatology run compared to the maximum surges and waves shows that the highest residuals are distributed over the North-West and South-West coast of England, while the highest Hs are distributed over the regions open to the Atlantic Ocean.' P14L188-190 and 'If the maximum waves, surges and tide (fig 5) could occur simultaneously, the highest TWLs would reach above 16 m in areas of the west of Ireland 13 m in areas of the Bristol channel.' P14L198-200. We hope this addresses your question.

5. P3L62 More information about the model setup is necessary: What are the initial and boundary conditions for the atmosphere and ocean? How often are boundary conditions updated? Is there only 1 domain (1.5km) or any nesting to reach such a high resolution (also depends on the resolution of input data)? What about model spin-up time? The first event (Xaver) occurred on Dec 5th/6th; so is the model already in balance or would more spin-up be necessary?

    **P3L79-89. We have now added more information on the model as well as more references that better explain the configuration of the model.**

    **To summarise, the initialisation and boundary conditions of the model is as follow:**

- **The Atmosphere model is initialised on 30 Oct 2013 by downscaled operational global UM analysis. The hourly lbcs come from a 24h global simulations initialised from archived operational global analysis and are updated daily. The grid resolution is variable, going from 1.5km in the inner (shelf) area of the domain and stretching out to 4km outer region to reduce impact of coarser-scale lateral boundaries (operational input data at 25km for dates of this study).**
- **The ocean model is initialised from long-run AMM15 hindcast. The daily lbcs come from an operational NATL12 1/12 North Atlantic model. The model grid is fixed resolution 1.5km grid (overlapping with the atmosphere grid over the shelf region).**
- **The wave model is initialised from a 10-days from rest wave-only spin-up. The hourly lbcs come from a global wave-only hindcast simulation.  The model grid varies from 3km in deep water (> 40 m) down to 1.5km in near-coastal areas.**

    **The atmosphere will spin up from the global initial state in about a day, the ocean and waves components are already spun up using the different initialisation strategies (i.e. long-run ocean hindcast and waves initialised 10-days prior to start time. The impact of coupling is also spun up as the simulation is initialised on the 30th October, more than one month before the first event. All files are obtained from/ saved to the Met Offices archives.**

6. P3L82 'higher temporal frequency than model output': where is this used in the study?

    **This is part of a sentence that refers to the methodology used in the model validation against observation. This line was moved in the appendix where we are giving information on the model validation, it does not belong in the main text. It has now been added to the captions of Table1 and Table2 that show the results from the comparison of model and observation.**

7. P13L115 Based on the Figures, it looks like surge increased much more than Hs; in the text, it says Hs is 138% higher on average, surge is 120% higher on average. Since the color bar (for

the positive percentages) seems to be the same, the much more reddish shades in Fig 4 conclude a much stronger average increase in surge. Please check and verify.

**Thank you for noticing this, it is an error. These numbers refer to a different experiment and not the data plotted on the figure. The correct numbers were replaced. The Hs is only 35% higher than on average, while the surge 106% higher than on average. Sorry about this.**

8.  P13L134 The curve representing simultaneous Hs and surges seem to be mostly influenced by surge; the effect of Hs seems negligible.

    **We have now specified this in P13L159.**

9.  P13 L145 A bit more interpretation of HP would de helpful: what does a value of 1.8m mean? It is calculated as SSH + 1/2Hs, where SSH is sea surface height including astronomical tides and surges. I'm having a problem how to interpret the HP value; maybe a bit more information should be added.

    **P13L171-P14L174 We have added the following line better describing what the Hp value represents.**

    **'This value, similarly, to the TWL, represents the simultaneous contribution to the water level from waves, surges and tides. However, operational thresholds are not based on just the maximum values of the TWL but are set considering extreme conditions, therefore the HP is calculated from the SSH (including astronomical tides and surges) summed to only half of the Hs rather than the maximum'.**

    **The average HP from winter 2013/4 is given in the paper, and what should be noted is the difference between the average value and the maximum values during that period. The limitation of the study is that with only 3 months of data we cannot show a true so climatology, with robust statistics derived from a long term run and what a typical value of the HP should be, however even during those 3 months it possible to see how drastically the TWL can be increased by an extreme event. Operationally speaking, whether that increase in HP is a threat or not for a specific coastal region will depend on other factors (such as further changes to the water level induced shallow water dynamics as the water propagates towards the coast, or local infrastructures and coastal defences). What we want to point out using the HP is how the water level can be affected by extreme events.**

    **We could have used the TWL values to show this instead, but the HP value seemed more appropriate as it is more like what flood warning thresholds would use as reference. Hopefully, this is clearer now.**

10. P14L156 'which of these are significant': have you applied any significance test? Otherwise, you should not use the word significant here?

    **Changed in P14L186. We have not applied significance tests, therefore this sentence was changed.**

11. P14L163 Based on your results (having the climatology and the 2013/2014) case: can you make any assumption about worst-case scenarios. & 15 Figure 5 How do Hs and surge

maxima look like in the climatology? Would probably also show the benefit of the high resolution in UK4.

**We have answered this together with point 4, see above.**

12. Figure 1 Is there a reason why you use mean atmospheric pressure as background? Why not topography/bathymetry?

    **Yes, the reason why we used the mean atmospheric pressure was to show how the pressure tended to be lower over the northwest region as most storms during that period travelled from the north Atlantic. We did not want to use the bathymetry as we though it could be misleading; while the bathymetry is used in the model processing steps, the SSH output is not given from the bathymetry itself but from an equipotential surface.**

13. Figure 2 The dark blue boxes highlight the periods, where HS and surge are both above the 90th percentile. Seems that the surge falls below the 90th percentile already at hour 35. Please verify.

    **This was a mistake in plotting the box, the new figure was corrected.**

14. Figure 4 'Note that the colour scales are different between figures 2-3.' Should be Figure 3 and 4, correct?

    **Yes, it should 3-4. This was corrected.**

15. Figure 5 How do Hs and surge maxima look like in the climatology? Would probably also show the benefit of the high resolution in UK4.

    **We have answered this together with point 4, see above.**

16. Figure 8 Add blank between HP and max in (b)

    **Done.**

P1L7 - Does the 7.7% refer to 90th %ile waves or surges or both?

**It refers to both. This is now specified in the text.**

**'High (>90ᵗʰ percentile) waves and high surges occurred simultaneously at any tidal stage, including high tide (7.7% of cases) [...]'**

Don't see any key words, but 'Ireland' might be a good one since its included here

**We will include this if keywords are asked by the journal. There do not seem to be a keyword space in the paper's template at the moment.**

Intro:

P1L18 - Worth mentioning that 2013/14 storms were particularly large and that the £592.1 million of damage is presumably a maximum rather than typical? I know this is made clear later on, but adding something like "during the exceptionally stormy winter of 2013/4..." would work here.

**This was added in the text.**

Also, were there flooding and damage reported then on the western Irish coast?
Seems as though there should have been with 14m waves!
Be good to add these cases in as well since Ireland is being resolved.

**We have added information and references with regards to Ireland and how it was affected by winter 2013/4. Extreme wave during this period did lead to flooding over the west coast and great geomorphological changes.**

**P1L19-22. 'In Ireland, the extreme conditions of that winter lead to high coastal erosion rates with extreme environmental impact (Sanchez-Arcilla et al. (2016),Cox et al. (2018), Janjic et al (2018)) and flooding, especially over the south, west, and north-west coasts (Met Eireann (2014), Thorne (2014));insurance claims and repairs resulting from the extreme weather damages amounted to more than €210 millions (Kandrot et al. (2016)).'**

P2L27 - "increase in the future"

**Done.**

P2L27 - Since mentioning other factors related to coastal flooding here, I think it would be good to mention the risk of compound flooding in estuaries from TWL and high river flows.

**Added, see P2L34-35.**

P2L32 - Please clarify what you mean by "residuals"

**Done. Now in P2L41.**

Methods:

P3L65 - Were rivers included in the model runs? Please clarify

**P3L76-77 This has now been clarified. The model uses climatological river inputs. A version of the UKC4 model coupled to the JULES river model does now exist, but this was still being implemented when we ran the simulation used in the study.**

P3L70 - I'd expect that the min 10m model coast would mean that your results are on the conservative side?

**We are not sure we understand this comment; if this refers to the fact that the model cannot resolve the shallow water processes, then yes, the results could be defined as conservative considering we cannot resolve near coast amplification. The assumption that the water is no shallower than 10 m means that waves do not experience bottom dissipation and are therefore bigger than they would be in reality, while for tide/surge the amplification is not large enough, so this component is underpredicted.**

Also in terms of timings relative to the tide, the timing of HW can shift through the intertidal, e.g in long estuaries, meaning that, for instance, an extreme Hs occuring an hour before HW at the coast of the UKC4 model might occur at HW further inshore. These points could be added.

**These points were added (P4L93-96) 'Shallow water dynamics as well as the tidal asymmetry and modulation of high waters in intertidal estuaries (Nidzieko2010) could lead to changes in the timing of waves, surges and tide inshore which cannot be represented here.'**

Results:

A lot of the text here is written in present tense, whereas past tense seems more appropriate to me.

**This was discussed but we decided to keep the present tense in the result section, as the tense refers to our present results.**

Fig2 - Nice figure.

**Thank you.**

Should the overlapped shading ending at 35 hrs actually end at 34 hrs, to reflect the end of teh surge >90th %ile?
**Yes, this was an error in the plot. It was corrected.**

Also might be clearer if the wave panel shades only waves >90%, the surge panel only surges >90%, and the overlap
between waves>90% and surges>90% is shown only in the tide panel below?

**Done.**

P13L115 - A note that Hs in Fig3 is reduced in a few spots in east/north.

**This was added to the figure 3 caption.**

Fig5 - could a more distinctive colour scale be used, and the outer black line removed?
This would make the figure clearer I think.
**This was discussed within the author team but, after making different versions of the figure, we feel that the changes do not improve the image. The colour scales chosen are equivalent to the cmocean deep and cmocean amp colourmaps, which are usually friendlier to colour blind readers.**

Also noticed that the 2013/14 surges on SE coast were 'normal' (Fig-4b), but the %ages in Fig5b in SE were high?
A breif discussion on this would be good.

**The 90th percentile of surges in Fig-4b from the CS3x model is normal, however the maximum surges are high. This shows both in the maximum surges from CS3x (not shown in the paper) as well as the UKC4 fig-5b. This region of the east coast was hit by one of the highest surges of the past 60 years during storm Xaver (5 December), which could explain the extreme increase in the maximum not reflected in the percentile.**

Discussion:
Generally excellent.

P15L212 - In the future...

**Done.**

Conclusion:

P16L232 - When 90th-%ile waves, surges and tides co-occurred, what/where was the flooding impact?

**As the paper deals with the high-water levels reached before inland propagation, we cannot draw direct conclusion on flooding. To answer your question, we have decided to upgrade part of the discussion, including adding reference to literature that deals with flooding during the period that we studied.**

**P14L193-L197 'During winter 2013/4, a sequence of extreme storms induced by an unusually strong North Atlantic jet stream (RMS2014) followed a more southerly path than usual (Thorne2014), causing extended flooding over the north of Scotland, west of Wales, west, south-west and south-east of England (Haigh2016, RMS2014, Thorne2014). The extent of storms over that season lead to question if the ongoing changes in the climate and typical storms conditions may also lead to changes in the spatial distribution and duration of floods (Thorne2014).'.**

**We have also added what/where the TWL was high at times in which 90th%ile wave, surges and high tide co-occurred P13L167-169 'During the period of coincidence between extreme waves, extreme surges, and high tides the regions showing the highest TWLs are over the west of Ireland, Scotland and in the Bristol Channel, followed by the north-west and south-east of England.'.**

**However, we have left the line in the conclusion unchanged as we cannot directly quantify the flooding impact in our own results.**

In the paper "Distribution of coastal high water level during extreme events around the UK and Irish Coasts" Julia Rulent and co-authors investigate the relative importance of waves, surges, tides, and their interaction to total water level elevations during the winter 2013/4 around the UK. The winter of 2013/4 was characterized by a number of heavy storms leading to significant flooding and damages. Here, the authors use a high resolution coupled ocean model to simulate the winter conditions and compare them relative to the climatological mean over 1979-2015. The authors find that the total water level distribution was heterogeneous along the coast and that the tide controls the timing of extreme surges in most cases (i.e. they usually occur during flood tide a few hours before high tide). The manuscript is very well written, the topic is an important one, and I have only a few comments and suggestions, none of which should prevent publication in NHESS.

Line 2: lead is a repetition to the first half of the sentence. Maybe better "trigger"

**Done.**

Lin 76-Line83: Could you please briefly elaborate on why the model performance is less good at locations such as Newlyn, Newport and Cromer? Are these very special locations that are difficult to model?

**There is a different plausible explanation for each TG as to why the bias was higher than at most stations.**

**In Newlyn there seem to be a lag between the TG and model reference level, with the model underestimating high tides and low tides peaks of a relatively constant value.**

**In Newport the TG seem to be within a harbour (see https://www.psmsl.org/data/obtaining/stations/351.php), while in Cromer the TG had problems during 2014 and was not levelled during that period (see https://www.psmsl.org/data/obtaining/stations/1632.php), which could explain part of the bias.**

**However, we do not know for sure why the difference is greater in these locations.**

Line 120: Am I reading the plot wrong? Isn't the overall maximum of Hs along the Shetland Islands?

**Shetland Islands also have some of the highest Hs, but the single overall maximum point is over the west of Ireland. As per your comment on Figure 5, the maximum points were marked on the plot.**

Caption Figure 2: (S)chematic

**Done.**

Figure 3: The text says that Hs have been 138% higher than the climatological mean, but the map looks like that it is far below 100 in most areas. Probably something wrong with the colorbar?

**Thank you for noticing this, it is an error that has now been corrected. These numbers refer to a different experiment and not the data plotted on the figure. The correct numbers were replaced.**

**The Hs is only 35% higher than on average, while the surge 106% higher than on average. Sorry about this.**

Figure 5: I think it might be helpful to mark the location of the overall maximum within each of the maps as the colorbar is not very easy to read.

**Done.**

Figure 8: (M)ean

**Done.**

Line 160: hazards(s)

**Done.**